# Application of a Nomogram Model in Predicting Postoperative Delirium Following Percutaneous Coronary Intervention

**DOI:** 10.3390/bioengineering12060637

**Published:** 2025-06-11

**Authors:** Yaxin Xiong, Ze Meng, Jiuyue Sun, Yucheng Qi, Kuo Wang, Ping Huang, Qiuyue Yang, Renliang Fan, Jiaman Guan, Mingyan Zhao, Xianglin Meng

**Affiliations:** 1Department of Critical Care Medicine, The First Affiliated Hospital of Harbin Medical University, Harbin Medical University, Harbin 150001, China; xiongyaxin99@outlook.com (Y.X.); 15603618617@163.com (Z.M.); 2024020752@hrbmu.edu.cn (J.S.); 2023021136@hrbmu.edu.cn (Y.Q.); wk592204029@163.com (K.W.); 2018183052@hrbmu.edu.cn (P.H.); 2019152247@hrbmu.edu.cn (Q.Y.); 2024020923@hrbmu.edu.cn (R.F.); 2024020926@hrbmu.edu.cn (J.G.); 2Heilongjiang Provincial Key Laboratory of Critical Care Medicine, The First Affiliated Hospital of Harbin Medical University, Harbin Medical University, Harbin 150001, China

**Keywords:** percutaneous coronary intervention, delirium, nomogram

## Abstract

**Background**: Postoperative delirium is associated with an increased number of different complications, such as prolonged hospital stay, long-term cognitive impairment, and increased mortality. Therefore, early prediction of delirium after percutaneous coronary intervention (PCI) is necessary, but currently, there is still a lack of reliable and effective prediction models for such patients. **Methods**: All data used in this study were derived from the MIMIC-IV database. Multivariable Cox regression was employed to analyze the data, and the performance of the newly developed nomogram was evaluated based on the area under the receiver operating characteristic curve (AUC). The clinical value of the prediction model was tested using decision curve analysis (DCA). **Results**: A total of 313 PCI patients in the intensive care unit (ICU) were included in the analysis, comprising 219 in the training cohort and 94 in the testing cohort. Twenty variables were selected for model development. Multivariable Cox regression revealed that benzodiazepine use, vasoactive drug therapy, age, white blood cell count (WBC), and serum potassium were independent risk factors for predicting the occurrence of delirium after PCI. The AUC values for predicting delirium occurrence in the training and validation cohorts were 0.771 and 0.743, respectively. **Conclusions**: This study has identified several important demographic and laboratory parameters associated with the occurrence of delirium after PCI, and used them to establish a more accurate and convenient nomogram model to predict the occurrence of postoperative delirium in such patients.

## 1. Introduction

Delirium is an acute neuropsychiatric syndrome characterized by fluctuating attention, environmental awareness, and cognitive/perceptual function, often accompanied by disturbances in the sleep–wake cycle, emotional lability, or hallucinations/delusions [1,2]. This syndrome not only serves as a marker of disease severity but is also significantly associated with increased inpatient and post-discharge mortality (hazard ratio 1.94–2.01) [3], prolonged hospital stays, and long-term cognitive decline [4]. Delirium also substantially increases patient costs, with studies indicating that hospitalized patients with delirium incur 55.4% higher inpatient expenditures and 100.7% higher 90-day expenditures compared to those without delirium.

Percutaneous coronary intervention (PCI), as the core method of revascularization for acute coronary syndrome, is widely used globally [5,6,7]. Despite advancements in professional knowledge, the development of new technologies, and improvements in anesthesia quality that have enhanced the safety of PCI, the incidence of postoperative delirium in patients undergoing PCI remains high [8,9,10].

Existing general delirium assessment tools cannot effectively capture the unique pathophysiological mechanisms associated with PCI, including hemodynamic fluctuations, coagulation abnormalities, and perioperative medication effects. A nomogram can integrate multiple predictive indicators, demonstrate the interrelationships between variables, and transform complex regression equations into intuitive graphics, making prediction results easier to read and facilitating rapid patient assessment by clinicians. This study aims to construct and validate a dedicated predictive model for PCI-related delirium using machine learning algorithms.

## 2. Methods

### 2.1. Data Source

Data were obtained from MIMIC-IV 3.0, a publicly accessible intensive care database [11]. It contains comprehensive information on patients admitted to the intensive care unit (ICU) at Beth Israel Deaconess Medical Center between 2008 and 2022. All personal information about patients in the database is de-identified, so informed consent was not required. The authors completed data research training through the Collaborating Institution Training Initiative to gain access to the database. All data were obtained from the official website of PhysioNet (https://mimic.physionet.org/, accessed on 1 May 2025).

### 2.2. Patient Population

A total of 872 patients met the screening criteria based on ICD codes, excluding those who underwent PCI before ICU admission, had a hospital stay of less than 24 h, or were under 18 years of age. Finally, 313 patients who met the inclusion criteria were selected. The procedure for data selection based on the above criteria is shown in Figure 1. We randomly selected 70% of the patients as the training cohort and used the remaining 30% as the validation cohort.

### 2.3. Data Extraction

Data were extracted using Structured Query Language. We not only extracted the patients’ basic information but also identified potential variables relevant to the assessment of postoperative delirium in patients undergoing PCI based on clinical experience and relevant literature. These variables included physiological indicators, blood test results, and medication usage. These data included the patient’s age, gender, admission mode, admission time, discharge time, death time, ICU admission time, and ICU discharge time. The first heart rate (HR), mean arterial pressure (MBP), blood oxygen saturation (SPO2), hemoglobin (Hb), platelets (PLTs), white blood cell count (WBC), blood urea nitrogen (BUN), serum creatinine (Cr), international normalized ratio (INR), activated partial thromboplastin time (APTT), bicarbonate, anion gap (AG), serum potassium, serum sodium, and blood glucose upon admission were also collected. Additionally, we recorded the use of benzodiazepines and vasoactive agents from admission to the preoperative period.

### 2.4. Outcome

The probability of delirium occurrence after PCI was determined as the primary outcome of this study, and the 28-day, 180-day, and 360-day mortality rates of the two groups of patients with and without delirium were determined as the secondary outcomes of this study.

### 2.5. Statistical Analysis

Details of missing data are summarized in Appendix A. There were no missing values for categorical variables, but some continuous variables had randomly missing data, with missing data for all variables being less than 20%. We used the mice package in R software to impute missing values using the predictive mean matching technique in the multiple imputation method [12], and visualized the distribution of data before and after imputation (Appendix A). We also performed descriptive statistical analysis of the data before and after imputation (Appendix A).

In this study, continuous variables with normal distribution were expressed as mean ± SD, while continuous variables with non-normal distribution were expressed as median and interquartile range [M (Q1, Q3)]. Categorical variables were presented as frequency and percentage. Comparisons of continuous variables were performed using the Wilcoxon rank-sum test, while comparisons of categorical variables were performed using Fisher’s exact test.

The death risks of patients in the no delirium and delirium present groups at 28 days, 180 days, and 360 days were analyzed using Kaplan–Meier (KM) survival analysis and the Cox proportional hazards regression model. Hazard ratios (HRs) and 95% confidence intervals (CIs) were calculated. To more accurately reveal the independent impact of grouping on outcomes, we also used a multivariable-adjusted Cox regression model to control for confounders. Covariates included gender, benzodiazepine use, vasoactive drug therapy, admission type, age, Hb, PLTs, WBC, BUN, Cr, INR, APTT, bicarbonate, MBP, HR, serum potassium, serum sodium, blood glucose, SP02, and AG. In the training cohort, the variance inflation factor (VIF) of all variables was calculated as an indicator of collinearity analysis. The results showed that the VIF values of all variables were less than 2, indicating that there was no significant collinearity problem among variables in our dataset. Multivariable logistic regression was used to analyze the included variables [13], and variables identified as significant were screened out to construct a nomogram [14]. The area under the receiver operating characteristic curve (AUC) was used to evaluate the predictive accuracy of the nomogram. The AUC ranges from 0 to 1, with a larger AUC value indicating more accurate prediction results. A calibration curve was plotted to evaluate the agreement between the predicted probability and the actual outcome, and decision curve analysis (DCA) was used to verify the clinical effectiveness of the prediction model [15].

All statistical analyses were performed using R software version 4.3.3 (R Foundation for Statistical Computing, Vienna, Austria). All tests were two-sided, and a *p*-value less than 0.05 was considered statistically significant.

## 3. Results

### 3.1. Baseline Characteristics of Patients

Of the 313 patients, 234 did not develop delirium within 7 days after surgery, and 79 developed delirium. As shown in Table 1, there were no significant differences between the two groups in terms of gender, age, admission type, Hb, PLTs, INR, APTT, bicarbonate, serum potassium, serum sodium, AG, or MBP. Compared with patients who did not develop delirium, patients who developed delirium had higher WBC, BUN, Cr, blood glucose, HR, and SPO2, and used more benzodiazepines and vasoactive drugs.

### 3.2. Survival Analysis of Patients

In this study, patients who developed delirium had a longer length of hospital stay, with a median length of stay of 15 days (IQR = 11–24 days). The in-hospital mortality rate and 28-day, 180-day, and 360-day mortality rates were also higher in patients who developed delirium, at 15 (19%), 15 (19%), 24 (30%), and 28 (35%), respectively, compared to patients who did not develop delirium, who had a median length of stay of 8.00 days (4.00, 13.00) and 28-day, 180-day, and 360-day mortality rates of 10 (4.3%), 18 (7.7%), 38 (16%), and 51 (22%), respectively (Appendix A). The Kaplan–Meier survival curve also showed that the survival rate was higher in patients who did not develop delirium than in those who did (Figure 2). Using univariate Cox regression analysis, it was found that the HRs (95% CI) for 28-day, 180-day, and 360-day mortality in patients who developed delirium, compared to those who did not, were 2.684 (1.353, 5.327), 2.103 (1.261, 3.506), and 1.854 (1.169, 2.940), respectively (Table 2). However, after controlling for potential confounders using multivariable Cox regression, only the 28-day mortality risk was significantly different between the two groups, with an HR (95% CI) of 2.833 (1.193, 6.728) (Table 2). This finding suggests that patients who developed delirium had a 1.833-fold increased risk of 28-day mortality compared to those who did not develop delirium.

### 3.3. Factors Associated with Delirium Occurrence in the Training Cohort

The 313 eligible PCI patients were randomly divided into a training cohort and a validation cohort at a ratio of 7:3, with 219 patients in the training cohort and 94 patients in the validation cohort. There were no significant differences in clinical pathological characteristic data between the two cohorts (Appendix A).

Multivariable Cox regression analysis identified five predictor variables that were significantly associated with delirium occurrence in patients: benzodiazepine use (OR = 2.486, *p* = 0.038), vasoactive drug therapy (OR = 3.815, *p* = 0.003), age (OR = 1.048, *p* = 0.008), WBC (OR = 1.048, *p* = 0.011), and serum potassium (OR = 0.462, *p* = 0.023) (Table 3). These variables were used to develop a visualized nomogram to predict the incidence of delirium after PCI in patients in the training cohort (Figure 3).

### 3.4. Model’s Predictive Performance and Clinical Utility

The model’s discriminative ability was evaluated using the receiver operating characteristic curve. The training cohort achieved an AUC of 0.771, while the validation cohort demonstrated an AUC of 0.743, indicating that the model can accurately distinguish patients who develop delirium from those who do not (Figure 4). The stability of the model’s discriminative power was assessed using the Bootstrap method (n = 1000 resampling iterations). Appendix A displays the distribution of AUC values, with a median of 0.762 and a 95% CI ranging from 0.737 to 0.773. No significant difference was observed compared to the original model, suggesting that the model’s predictive capability remains robust against sample fluctuations.

To evaluate the calibration of the multivariate logistic regression model, calibration curves were plotted for both the training and validation cohorts. The x-axis represents the predicted risk of delirium occurrence, while the y-axis denotes the actual observed risk. The diagonal gray line represents perfect prediction in an ideal model, and the dashed line indicates the performance of the nomogram. The closer the dashed line is to the diagonal, the better the prediction. The calibration curves of the nomogram closely align with the ideal curve, indicating a high degree of consistency between predicted probabilities and actual outcomes (Figure 5). The Hosmer–Lemeshow test further confirmed the model’s excellent calibration (training set: χ^2^ = 9.204, *p* = 0.419; validation set: χ^2^ = 8.750, *p* = 0.461) (Figure 5).

In the DCA, the x-axis represents the threshold probability, and the y-axis denotes the net benefit. The horizontal one indicates that all samples are negative and all are not treated, with a net benefit of 0. The oblique one indicates that all samples are positive. The net benefit is a backslash with a negative slope. The DCA results reveal that when the threshold probability ranges between 9% and 62% (training cohort) and 11% and 66% (validation cohort), the nomogram is effectively useful for predicting the occurrence of postoperative delirium after PCI (Figure 6).

## 4. Discussion

Postoperative delirium was associated with an increased number of different complications. Similar to previous research findings [9,16], our study found that patients who developed delirium after PCI had longer hospital stays and increased mortality rates. Multivariable Cox regression showed that delirium patients had a significantly increased risk of 28-day mortality (HR = 2.833), but this association disappeared in the long term (180 days/360 days). This result differs from the significant impact of delirium on both short-term and long-term mortality in patients after other major surgeries [17,18]. This study suggests that postoperative delirium may only be an independent risk factor for increased short-term mortality in PCI patients, and its mechanism may be directly related to acute complications (such as aspiration, arrhythmia) induced by delirium [19,20], while long-term mortality is more likely driven by underlying diseases (such as heart failure, chronic kidney disease). This finding emphasizes the importance of early postoperative delirium monitoring. This study constructed a risk prediction model for postoperative delirium in PCI patients based on age, WBC and serum potassium levels at admission, and preoperative use of benzodiazepines and vasoactive drugs, providing healthcare professionals with an accurate and objective assessment tool.

This study found that age, as an independent risk factor for delirium [21,22], is also important in predicting delirium after PCI. In elderly patients, vascular stiffness and endothelial dysfunction lead to impaired cerebral autoregulation reserve capacity [23,24]. Studies have shown that intraoperative blood pressure fluctuations exceeding the range of cerebral autoregulation can cause cerebral hypoperfusion, disrupt the match between neuronal metabolic demand and blood flow supply, and increase the risk of postoperative delirium [25]. Secondly, the blood–brain barrier permeability in elderly patients has already increased due to aging or neurodegenerative diseases (such as Alzheimer’s disease) before surgery. Surgical stress (such as the release of inflammatory factors and ischemia-reperfusion injury) further damages the blood–brain barrier, allowing inflammatory mediators and neurotoxic substances to infiltrate the brain parenchyma, triggering neuroinflammation and synaptic dysfunction [26,27,28]. Finally, the clearance rate of anesthetic drugs is reduced in the elderly population, which may prolong the central inhibitory effects of the drugs and exacerbate postoperative cognitive impairment [29]. Due to the limited sample size, our study did not specifically investigate the incidence of postoperative delirium in younger patients (particularly premenopausal women) following PCI. The risk of delirium in younger females may differ due to the potential neuroprotective effects of estrogen [30]. Future investigations are warranted to elucidate the underlying mechanisms of postoperative delirium in this specific population. The preoperative use of vasoactive drugs showed the most prominent predictive efficacy in the prediction model of delirium after PCI. Some vasoactive drugs (such as dopamine and epinephrine) may induce delirium by directly interfering with the dopamine/acetylcholine balance, activating the adrenergic system, or inducing neuroinflammation [31,32,33,34,35]. Secondly, the potent vasoconstriction effect of phenylephrine may reduce cerebral perfusion and is positively correlated with postoperative delirium [36,37]. Finally, the use of vasoactive drugs often serves as a surrogate indicator of hemodynamic instability, suggesting that patients have more severe circulatory compensation disorders, resulting in insufficient cerebral blood and oxygen supply and leading to delirium [38,39]. Future research needs to further explore the gradient effects of the type, dose, and duration of vasoactive drugs on delirium risk and develop dynamic prediction models based on real-time hemodynamic data to achieve personalized medication guidance. Elevated WBC is a sensitive indicator of systemic inflammation. PCI surgical trauma, ischemia–reperfusion injury, or postoperative infection can activate the immune system, leading to the massive release of inflammatory cells such as neutrophils and monocytes, and promoting the elevation of pro-inflammatory cytokines (such as IL-6 and TNF-α) [40,41,42]. Studies have shown that postoperative systemic inflammation indexes (such as NLR and PLR) are significantly correlated with the incidence of delirium [43,44], suggesting that systemic inflammatory status is an important trigger for delirium. Some studies have pointed out that perioperative decreased serum potassium levels are independent risk factors for postoperative delirium, and hypokalemia, along with hyponatremia, low CRH, and hyperglycemia, constitutes a metabolic risk profile for delirium. This may be related to the impact of electrolyte imbalance on neuronal electrical activity and neurotransmitter release [45]. Ketema et al.’s study also confirmed that electrolyte disturbances are significant risk factors for delirium after acute stroke [46], suggesting that hypokalemia may induce delirium through direct or indirect neurometabolic disorders. This is consistent with our findings, and future experiments are needed to further explore the potential value of targeted potassium supplementation interventions in preventing delirium. In this study, preoperative benzodiazepine use emerged as a significant predictor in the prediction of postoperative delirium following PCI. Benzodiazepines, by enhancing GABAergic inhibition, may excessively inhibit thalamic or cortical excitatory neurons, leading to decreased or fluctuating levels of consciousness [47,48]. Notably, certain benzodiazepines may influence WBC counts [49] or indirectly modulate neuronal excitability through potassium channel activation [50,51]. These mechanistic associations could potentially introduce confounding effects between variables. Future investigations are warranted to rigorously validate the dose–response relationships among pharmacodynamic variations, metabolic disparities, and delirium onset in this clinical context.

The prediction model developed in this study has good clinical application value. First, the variables in the model are all basic patient information and routine clinical test indicators, which are easy to obtain and monitor. Secondly, the model has good predictive performance, with AUC values of 0.771 and 0.743 in the training and validation cohorts, respectively. Finally, the nomogram prediction model visually integrates multivariate information, intuitively quantifies the independent contributions of each factor to the outcome, and achieves personalized probability prediction. Therefore, in clinical practice, doctors can use this model to conduct early risk assessment of patients, formulate more reasonable treatment plans and prognosis management strategies, and prevent the occurrence of delirium after PCI.

We developed a comprehensive nomogram to predict postoperative delirium in PCI patients admitted to the ICU, providing guidance for the clinical management of such patients. However, our study has several limitations. First, our study used single-center data from the United States, which may limit the applicability of our findings to patient populations from different regions, races, and socioeconomic backgrounds, and those with varying medical levels. Second, although we screened most variables, including basic patient information, physiological indicators, blood tests, and admission assessments, for analysis, there may still be potential predictor variables that were not included, which could lead to selection bias and affect the research results. Finally, our model only underwent internal validation, and future research needs to perform external validation using independent datasets to further confirm the performance and accuracy of this novel nomogram.

## 5. Conclusions

In conclusion, we developed, for the first time, a nomogram prediction model which has good predictive value for the occurrence of delirium after PCI. Future research should further validate this model in different cohorts.

## Figures and Tables

**Figure 1 bioengineering-12-00637-f001:**
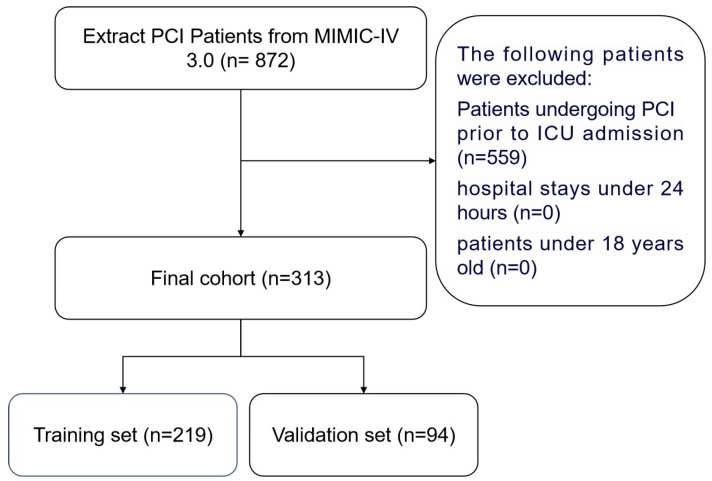
Flowchart of the study. PCI, percutaneous coronary intervention; MIMIC IV, Multiparameter Intelligent Monitoring in Intensive Care Database IV.

**Figure 2 bioengineering-12-00637-f002:**
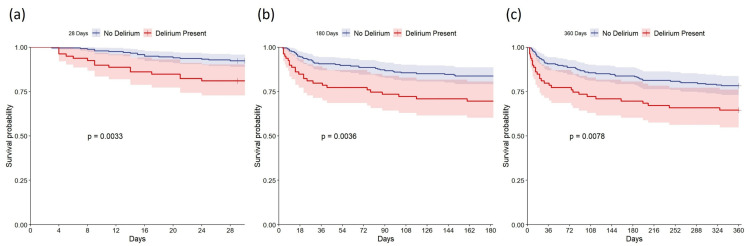
Kaplan-–Meier survival curves comparing patients with no delirium vs. those with delirium. (**a**): 28-day survival rate; (**b**): 180-day survival rate; (**c**): 360-day survival rate.

**Figure 3 bioengineering-12-00637-f003:**
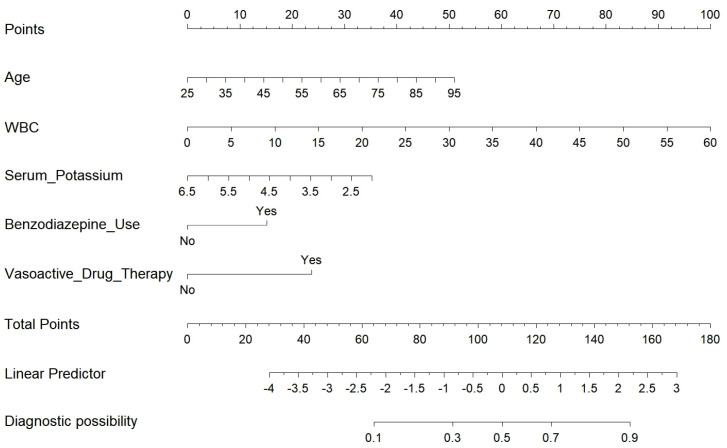
Nomogram for predicting postoperative delirium following percutaneous coronary intervention.

**Figure 4 bioengineering-12-00637-f004:**
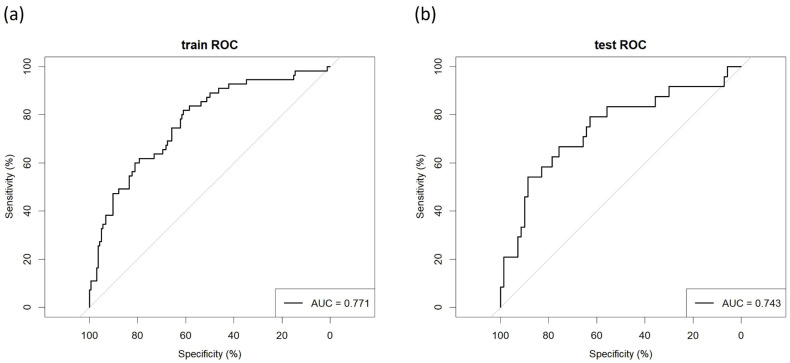
Area under the ROC curves (AUC) of training cohort (**a**) and validation cohort (**b**).

**Figure 5 bioengineering-12-00637-f005:**
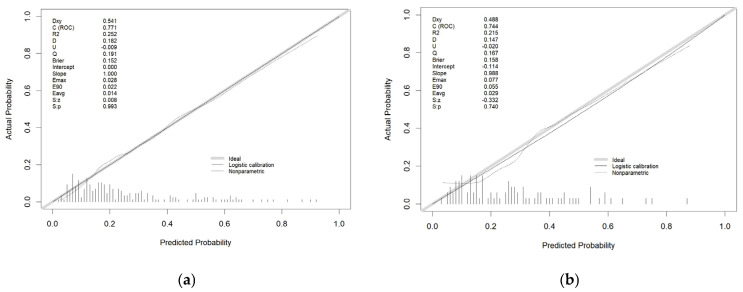
Calibration plots showing the relationship between the predicted probabilities based on the nomogram and actual values of the training cohort (**a**) and validation cohort (**b**).

**Figure 6 bioengineering-12-00637-f006:**
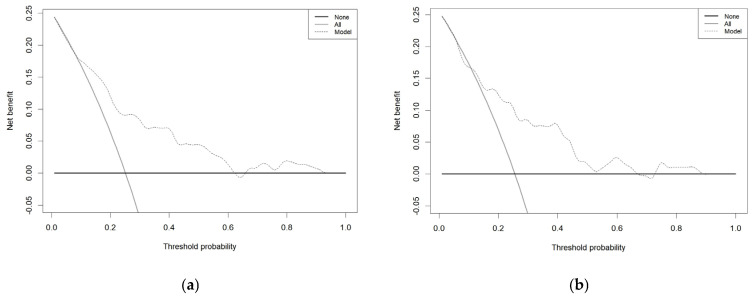
Decision curve analysis. The abscissa is the threshold probability, and the ordinate is the net benefit rate. The horizontal one indicates that all samples are negative and all are not treated, with a net benefit of 0. The oblique one indicates that all samples are positive. The net benefit is a backslash with a negative slope. Training cohort (**a**) and validation cohort (**b**).

**Table 1 bioengineering-12-00637-t001:** Baseline characteristics and prognosis of patients after PCI.

Characteristic	No Delirium, N = 234	Delirium Present, N = 79	*p*-Value
Gender			0.883
Male	138 (59%)	48 (61%)	
Female	96 (41%)	31 (39%)	
Age	71.00 (62.00, 80.00)	73.00 (62.00, 82.00)	0.452
Admission Type			0.799
Emergency	109 (47%)	40 (51%)	
Urgent	97 (41%)	31 (39%)	
Other	28 (12%)	8 (10%)	
Hb	11.80 (10.20, 13.30)	11.30 (9.35, 13.15)	0.226
PLTs	203.00 (153.25, 264.75)	215.00 (158.00, 291.00)	0.302
WBC	9.85 (7.30, 13.50)	12.20 (8.30, 16.75)	<0.001
BUN	23.00 (16.00, 30.00)	27.00 (18.50, 44.50)	0.007
Cr	1.10 (0.80, 1.50)	1.40 (0.90, 1.95)	0.019
INR	1.20 (1.10, 1.40)	1.20 (1.10, 1.45)	0.128
APTT	35.45 (29.33, 59.93)	36.40 (29.55, 61.25)	0.718
Bicarbonate	24.00 (21.00, 26.00)	23.00 (19.00, 26.00)	0.124
Serum Potassium	4.10 (3.80, 4.70)	4.30 (3.65, 4.70)	0.590
Serum Sodium	138.00 (135.25, 140.00)	138.00 (135.00, 141.00)	0.785
Blood Glucose	135.00 (106.00, 179.75)	154.00 (122.00, 201.50)	0.032
AG	15.00 (13.00, 18.00)	16.00 (14.00, 19.00)	0.104
MBP	80.00 (70.00, 93.00)	81.00 (71.00, 100.00)	0.208
HR	85.50 (71.00, 102.00)	94.00 (75.00, 111.00)	0.037
SP02	97.00 (94.00, 99.00)	98.00 (95.00, 100.00)	0.045
Benzodiazepine Use	75 (32%)	44 (56%)	<0.001
Vasoactive Drug Therapy	44 (19%)	45 (57%)	<0.001
Hospital LOS	8.00 (4.00, 13.00)	15.00 (11.00, 24.00)	<0.001
Hospital Mortality	10 (4.3%)	15 (19%)	<0.001
28-day mortality	18 (7.7%)	15 (19%)	0.009
180-day mortality	38 (16%)	24 (30%)	0.010
360-day mortality	51 (22%)	28 (35%)	0.024

**Table 2 bioengineering-12-00637-t002:** HRs (95% CIs) for all-cause mortality.

	No Delirium (*n* = 234)	Delirium Present (*n* = 79)	
		HR (95%CI)	*p*-Value
28-day mortality			
Unadjusted	Reference	2.684 (1.353, 5.327)	0.005
Adjusted	Reference	2.833 (1.193, 6.728)	0.018
180-day mortality			
Unadjusted	Reference	2.103 (1.261, 3.506)	0.004
Adjusted	Reference	1.599 (0.841, 3.040)	0.153
360-day mortality			
Unadjusted	Reference	1.854 (1.169, 2.940)	0.009
Adjusted	Reference	1.640 (0.931, 2.888)	0.087

**Table 3 bioengineering-12-00637-t003:** Multivariate Cox regression analysis.

Variables	OR	95% CI	*p*-Value
Gender			
Male	Reference	—	—
Female	0.784	(0.331–1.808)	0.573
Age	1.048	(1.013–1.086)	0.008
Admission Type			
Emergency	Reference	—	—
Urgent	1.450	(0.632–3.388)	0.382
Other	1.640	(0.461–5.460)	0.427
Hb	1.153	(0.941–1.423)	0.175
PLTs	1.003	(0.999–1.007)	0.166
WBC	1.071	(1.019–1.136)	0.011
BUN	1.021	(0.997–1.046)	0.090
Cr	1.116	(0.739–1.612)	0.576
INR	0.923	(0.557–1.444)	0.738
APTT	0.993	(0.979–1.006)	0.302
Bicarbonate	0.947	(0.830–1.074)	0.407
Serum Potassium	0.462	(0.231–0.881)	0.023
Serum Sodium	1.047	(0.960–1.155)	0.330
Blood Glucose	1.003	(0.997–1.008)	0.364
AG	0.939	(0.804–1.093)	0.423
MBP	1.023	(1.000–1.048)	0.053
HR	1.012	(0.993–1.032)	0.211
SP02	1.063	(0.963–1.190)	0.253
Benzodiazepine Use			
NO	Reference	—	—
YES	2.486	(1.055–5.990)	0.038
Vasoactive Drug Therapy			
NO	Reference	—	—
YES	3.815	(1.575–9.479)	0.003

## Data Availability

The raw data supporting the findings of this study were obtained from the MIMIC-IV-3.0 database, which is publicly available through the PhysioNet platform at https://physionet.org/content/mimiciv/3.0/ (accessed on 24 January 2024).

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
