# Peer review of "Application of a Nomogram Model in Predicting Postoperative Delirium Following Percutaneous Coronary Intervention"

_bioengineering, 2025, doi:10.3390/bioengineering12060637_

Round 1
Reviewer 1 Report
Comments and Suggestions for Authors
The team of Authors from Harbin Medical University has conducted a clinical data study using multivariable Cox regression analysis to identify important factors that are statistically associated with the occurrence of delirium after percutaneous coronary intervention (PCI). As a result of the analysis of 313 patients' data set (including 219 patients in the training cohort and 94 in testing cohort) the team of investigators found multiple variables (benzodiazepine use, vasoactive drug therapy, age, WBC, and serum potassium) to were independent risk factors for predicting the occurrence of delirium after PCI convenient. The authors developed a nomogram model to predict the occurrence of postoperative delirium in a clinical setting, which makes this work particularly interesting and valuable.
My only suggestion for improvement is to add a section that includes a list of all used abbreviations. Additionally, authors need to address the fact that certain benzodiazepine drugs could affect WBC. For example, alprazolam (Xanax) and clonazepam (Klonopin) have been linked to potential immune system impacts, including altered white blood cell counts (WBCs). Specifically, alprazolam has been shown to increase neutrophil counts and decrease lymphocyte counts. Additionally, some benzodiazepines have been found to activate Kv7.1 potassium channels, leading to a shortening of action potential duration. Others can modulate calcium channels, which can also affect neuronal excitability. Therefore, even if the statistical analysis might suggest that benzos, WBC, and potassium levels might be independent variables, there might be some confounding effects and interdependence among these factors that might need further validation and investigation. Additionally, the average age of the cohort seems to be around 70 years old (between 62 to 80), and it is not clear what the effect might be of younger ages, especially in females who are not postmenopausal. It might be worth adding a little more detailed discussion on the occurrence and incidence of delirium in PCI patients of younger age. The average age for women under 45 undergoing percutaneous coronary intervention (PCI) due to coronary artery disease is generally around 40 years old. Studies have shown that the average age of female patients in this age group undergoing PCI is typically between 39 and 41 years.Author Response
请参阅附件。

Reviewer 2 Report
Comments and Suggestions for Authors
I was glad to read the presented article, devoted to the very important topic of assessing various factors for the risk of postoperative delirium and its possible consequences.
In section 3.4. after a short text there is a series of graphs without text accompaniment. Perhaps it would be better to supplement the existing text a little and break it into meaningful paragraphs concerning each graph, so that all of them could be accompanied by a relevant description.
The authors themselves wrote that perhaps not all potentially important factors were taken into account in this study. Nevertheless, some obvious and common parameters, such as the presence of arrhythmia, signs of kidney and liver pathologies, body mass index, probably should have been included in the consideration (if there was access to them).
In the abbreviations section, the abbreviations are deciphered (even if it was previously done in the main text), which is quite convenient for the reader. However, not all abbreviations are given (see p. 3), and nowhere in the text the abbreviation PT (prothrombin?) from Table S2 was deciphered.
Did the authors evaluate the benefit (for the final results) of excluding some examples that differ greatly (in terms of statistics) in the training model?
In the section of supplementary materials, in Figure S1 the graphs are very small in scale and are difficult to read even when enlarged in .pdf format, and are completely unreadable in the printed version.
These comments do not change my positive opinion of this work, so I believe that it can be accepted for publication in its current version or after minor edits (including those I have noted).
Reviewer 3 Report
Comments and Suggestions for Authors
This study addresses an important clinical issue by developing a nomogram to predict postoperative delirium in ICU patients undergoing percutaneous coronary intervention (PCI) using data from the MIMIC-IV database. A total of 313 PCI patients in the intensive care unit (ICU) were included in the analysis, comprising 219 in the training cohort and 94 in the testing cohort. Overall, the study presents promising findings and will be attractive to the fields. However there are still some minor issues that authors need to solve.
1. What specific inclusion and exclusion criteria were used to select the 313 PCI patients? Were any patients excluded due to missing data or pre-existing cognitive disorders?
2. How were the 20 variables initially selected—through literature review, clinical judgment, or data-driven methods?
3. Given the AUCs of 0.771 (train) and 0.743 (test), how do the authors assess the clinical utility of the model, especially given potential overfitting in small sample sizes?
4. The words and legends in Fig. S1 are too small to see. Please divide it into several figures or make it bigger.
